# Microbiological and Ergonomic Effects of Three Prototypes of a Device to Reduce Aerosol Dispersion in Dental Care During the COVID-19 Pandemic: A Randomized Controlled Clinical Trial

**DOI:** 10.3390/dj13020054

**Published:** 2025-01-26

**Authors:** Camila N. Baldasso, Ruy Teichert-Filho, Daniel R. Marinowic, Maria M. Campos, Maximiliano S. Gomes

**Affiliations:** 1Post-Graduate Program in Dentistry, School of Health and Life Sciences, Pontifícia Universidade Católica do Rio Grande do Sul, Porto Alegre 90619-900, Brazil; camilanbaldasso@gmail.com (C.N.B.); ruyteichert@gmail.com (R.T.-F.); maria.campos@pucrs.br (M.M.C.); 2Medical and Dental Center, Military Police of Rio Grande do Sul, Porto Alegre 90010-190, Brazil; 3Brain Institute (InsCer), Pontifícia Universidade Católica do Rio Grande do Sul, Porto Alegre 90619-900, Brazil; daniel.marinowic@pucrs.br; 4Post-Graduate Program in Medicine and Health Sciences, School of Medicine, Pontifícia Universidade Católica do Rio Grande do Sul, Porto Alegre 90619-900, Brazil

**Keywords:** dental care, infection prevention and control, personal protective equipment, public health

## Abstract

**Objectives**: This randomized clinical trial evaluated the microbiological efficacy and the ergonomic impact of three prototypes of a device to reduce aerosol dispersion during dental procedures. **Methods**: Sixty patients undergoing dental care using high-speed turbines and/or ultrasonic tips were randomly assigned to 4 groups (*n* = 15): CG: control group, with standard personal protective equipment (PPE); G1: PPE + acrylic device (AD) with aspiration; G2: PPE + AD without aspiration; and G3: PPE + polyvinyl chloride device. The device prototypes consisted of a rigid translucent acrylic structure (G1 and G2), or a rigid PVC tube structure surrounded by layers of translucent flexible PVC films (G3), adjusted to the dental chair, involving the patient’s head, neck and chest. The main outcome was the microbiological analysis (mean Δ of CFU at 10 different sites), and the secondary outcome was the ergonomic evaluation (questionnaire to dentists and patients). **Results**: The final sample comprised 59 participants (mean age 38.6 ± 11.4 years, 55.2% male). The overall mean time for dental procedures was 32.4 ± 16.9 min, with no differences between groups (*p* = 0.348). Microbiological analyses showed that the use of the device significantly reduced contamination in the light reflector (01.46 ± 4.43 ΔCFU in G2 vs. 19.25 ± 36.50 ΔCFU in CG; *p* = 0.028), apron (09.11 ± 12.05 ΔCFU in G3 vs. 21.14 ± 43.41 ΔCFU in GC; *p* = 0.044), and face shield (08.80 ± 32.70 ΔCFU in G1 vs. 56.78 ± 76.64 ΔCFU in the GC; *p* = 0.017). The device was well accepted by patients and increased the dentists‘ perception of safety and protection (*p* < 0.001), but significantly decreased ergonomics related to the clinical view, space, agility and access to the patient, and ease of performing procedures (*p* < 0.001). **Conclusions**: The tested device can be an additional tool for infection prevention and control in dentistry, not only during the COVID-19 pandemic, but also for the control of future infectious diseases and epidemics.

## 1. Introduction

The COVID-19 pandemic was one of the most serious global health challenges in human history [1,2] with a severe impact on health services and profound economic, educational, and social consequences [3,4]. Dental services were severely affected, and several strategies were implemented to improve infection prevention and control, including changes in routine dental care protocols, patient scheduling, and financial management [5,6].

The aerosol generated during medical and dental procedures is a major challenge because it can contain many pathogenic microorganisms, including bacteria and viruses [7]. By definition, an aerosol is a suspension of fine solid particles or liquid droplets in a gas. Aerosols can vary greatly in size, with the term often used to describe particles smaller than 100 µm, which can remain airborne for extended periods of time [8]. Aerosol particles <10 µm can penetrate into the lower respiratory tract, facilitating the transmission of respiratory pathogens such as the severe acute respiratory syndrome coronavirus 2 (SARS-CoV-2). Some protective devices have been previously proposed, mainly in the medical field, to reduce aerosol dispersion [9,10,11,12,13,14].

In dentistry, the high load of microorganisms in the mouth and the close contact between patients and dental professionals represent a common high-risk condition. SARS-CoV-2 RNA has been frequently identified in supragingival and subgingival biofilms, independent of a patients’ periodontal status and systemic viral load [15]. During the pandemic period, dental practice required extreme additional and innovative complementary resources to limit the SARS-CoV-2 viral spread. Some initiatives have been proposed to reduce contamination during dental care, such as various devices to prevent aerosol dispersion [16,17,18,19] or procedures that do not produce aerosol [20].

To date, the microbiological efficacy of existing devices and their ergonomic impact in dental care have not been sufficiently tested in clinical settings. Most previous studies were conducted under laboratory conditions and used simulated procedures to analyze the devices, which may not reflect real-life dental care situations. Therefore, the aim of this randomized clinical trial was to evaluate the microbiological efficacy and ergonomic impact of three prototypes of a device to reduce aerosol dispersion in dental care during the COVID-19 pandemic.

## 2. Materials and Methods

### 2.1. Trial Design and Ethical Considerations

This randomized, controlled, open-label, single-center, parallel clinical trial was conducted at the Medical and Dental Center of the Military Police of Rio Grande do Sul (CMOBM), Brazil, from August 2020 to July 2021. This study was approved by the Ethics Committee of the Pontifical Catholic University of Rio Grande do Sul (CAEE # 30717820.1.0000.5336, 12nd May 2020) and by the Research Institute of the Military Police of Rio Grande do Sul (PROA #20/1203-0008003-4, 6th May 2020). This study was registered in the Brazilian Registry of Clinical Trials (RBR-9rnkbs) with Universal Trial Number (UTN) U1111-1258-5375, 21th September 2020, and followed the guidelines described in the CONSORT 2010 (Consolidated Standards of Reporting Trials) statement on clinical trials. All participants provided an informed consent form.

### 2.2. Study Population, Recruitment, Randomization and Allocation

Consecutive patients seeking dental care at the Medical and Dental Center of the Military Police of Rio Grande do Sul, Brazil, were invited to participate in the study. The inclusion criteria were: (1) individuals of any sex, over 18 years of age; (2) need for dental treatment using high-speed turbines and/or ultrasonic tips; and (3) voluntary agreement to participate in the study. Exclusion criteria were: (1) current antibiotic therapy or antibiotic use in the last 6 months; (2) need for dental care without an indication for procedures requiring the use of high-speed turbines (KaVo, Joinville-SC, Brazil) and/or ultrasonic tips (Gnatus, Barretos-SP, Brazil); and (3) presence of signs and symptoms compatible with SARS-CoV-2 at the time of the dental visit.

The sample-size calculation was carried out to detect minimal differences between the experimental groups of 8 × 10^4^ CFU, with a standard deviation of 7 × 10^4^ CFU, based on the results of a previous study [21]. Using a 5% significance level and 80% power, the sample-size calculation resulted in 13 participants per group. Considering a potential drop-off rate of 15%, the final sample resulted in 15 participants per group, with a total of 60 participants.

Patients were randomly assigned to one of the four experimental groups using an electronic algorithm (random.org). The sequence of sample allocation followed a simple random sequence until a given group reached the sample of 15 individuals. Figure 1 shows the CONSORT flowchart of sample selection and allocation.

The four experimental groups were:-Control Group (CG): regular dental visit with the operator using standard personal protective equipment (PPE), including surgical gloves, apron, mask, cap, and face shield;-Group 1 (G1): dental visit with the operator wearing the same standard PPE as described in CG, with the addition of an acrylic device with an active aspiration system. This device has been described in detail previously [16];-Group 2 (G2): dental visit with the operator using the same standard PPE described in CG, with the inclusion of the same acrylic device used in G1, but without the aspiration system;-Group 3 (G3): dental visit with the operator using the same standard PPE described in CG, with the inclusion of a simplified polyvinyl chloride (PVC) device. This device consisted of a rigid PVC tube (Tigre, Joinville-SC, Brazil) structure surrounded by layers of translucent flexible PVC films (Ultraplast, Maceio-AL, Brazil). As in G1 and G2, the device used in G3 was adapted to the dentist’s chair, involving the patient’s head and chest.

In G1, G2, and G3, the operators and dental assistants worked through small holes in the acrylic structure or in the PVC film, as described previously [16]. Figure 2 shows images of dental procedures performed using the prototypes in G1, G2, and G3.

All dental visits were performed in a cleaned and disinfected dental office. Routine surface disinfection included the use of detergent, 70% alcohol, and 2% sodium hypochlorite. Natural ventilation was used in all dental offices according to routines established by the CMOBM dental service during the COVID-19 pandemic. Prior to dental visits, microbial samples were collected by swabbing from the following locations: light reflector, dental chair headrest, auxiliary clinical table, dentist apron, dentist mask, dentist face shield, dentist cap, protective bib on the patient’s chest, office bench, and operating room wall. Swab specimens were collected from the same locations immediately after completion of the dental procedures.

Swab samples were transferred to tubes containing sterile saline and transported for microbiological processing. After preparation of serial dilutions, viable bacteria were cultured in Petri plates containing blood-agar medium (Blood-Agar-TSA-20mL-PL 90X15-PC 10PL, Laborclin, Pinhais-PR, Brazil) and incubated at 37 °C for 24 h. At the end of the incubation period, the number of colony forming units (CFU) was counted and evaluated according to classical microbiological methods.

In addition, participants (both patients and dentists) underwent COVID-19 testing using nasopharyngeal and oropharyngeal swab samples from 3 to 6 h after sample collection using an SV-Total RNA kit (Promega, Madison, WI, USA), according to the manufacturer’s recommendations. For amplification, the protocol established by the Centers for Disease Control and Prevention (CDC) was used through the AgPath-IDT master mix (Applied Biosystems, Thermo Fisher Scientific, Grand Island, New York, NY, USA) and probes complementary to the regions of the N1 and N2 genes of SARS-CoV-2, in addition to the endogenous control Ribonuclease P (RP).

### 2.3. Ergonomic Analysis

Semi-structured questionnaires were administered to all participants, both patients and dentists, to capture their perceptions of the ergonomic impact of the different device prototypes. Dentists answered a questionnaire with six questions related to the following domains: sense of safety and protection (Q1: During the dental procedure, how was your feeling of safety and protection in relation to the risks of contamination?); clinical visualization (Q2: During the dental procedure, how was your clinical visualization to perform the procedure?); physical access to the patient (Q3: During the dental procedure, how was your feeling of limitation regarding the physical access to the patient?); ease of performing procedures (Q4: During the dental procedure, how was your feeling regarding the ease to execute the surgical/operative procedures?); agility of movement (Q5: During the dental procedure, how was your feeling of agility in moving and changing your working position?); and space to work (Q6: During the dental procedure, how was your feeling in relation to the space to work?). The patient questionnaire had a single question: “How was your experience using the device during the dental visit?” The answers to all these questions were structured as a five-point Likert scale (1: very bad; 2: bad; 3: regular; 4: good; 5: very good). The questionnaires for both patients and dentists included a final open-ended question to allow for free opinions and suggestions regarding the use of the device.

### 2.4. Outcome Variables

The main outcome was the mean Δ of CFU identified at each collected point, calculated by the difference of CFU before and after the dental visit. In addition, the overall mean Δ of CFU was calculated considering the mean of all 10 collected points.

Secondary outcomes included ergonomic evaluation, analyzed separately for patients and dentists. Ergonomic outcomes were expressed as the distribution and mean for each Likert-scale question. Qualitative analysis was applied to the open-ended questions.

### 2.5. Other Variables

Additional variables were collected through a questionnaire administered to patients and dentists. The following variables were collected from patients: sociodemographic variables (age, sex, and education level) and medical history (weight, height, diabetes, smoking, hypertension, and cardiovascular disease). Variables related to dentists included age, sex, and dental specialty. The following dental visit-related variables were also collected: type of dental treatment (restorative, coronal access, and prophylaxis), region of dental treatment (upper incisors or canines, lower incisors or canines, upper bicuspid or molar, upper bicuspid or molar, and all teeth), and duration of dental visit (measured in minutes).

### 2.6. Statistical Analysis

The raw data of the study, including all variables analysed, are available in the Appendix A. The data were first described by descriptive statistics (frequencies, means, and standard deviation), followed by the Kolmogorov–Smirnov normality test. Differences between groups with respect to sociodemographic and medical characteristics of the participants and variables related to dental procedures were estimated using the Kruskal–Wallis test for independent samples, with pairwise comparison (for continuous nonparametric variables) or the Chi-square test (for categorical variables).

For the microbiological data, the mean Δ of CFU at each collection point was calculated by the difference of CFU before and after the dental visit. In addition, the overall mean Δ of CFU was calculated considering the mean of all 10 collected points. Differences between groups were estimated using the Kruskal–Wallis test for independent samples with pairwise comparison (continuous nonparametric variables). For ergonomic data, differences between groups were estimated by the Kruskal–Wallis test for independent samples with pairwise comparison (for continuous nonparametric variables) or the Chi-square test (for categorical variables). Statistical analyses were performed using the Statistical Package for Social Science software (IBM SPSS v.20.0 for Mac; SPSS Inc., Chicago, IL, USA). For all analyses, the null hypothesis was rejected at a 5% significance level.

## 3. Results

### 3.1. Characteristics of the Sample

The sociodemographic and medical characteristics of the participants, as well as the variables related to the operative procedures in the different groups, are expressed in Table 1. One patient in the CG withdrew from participation in the study, and the final sample consisted of 59 individuals (mean age 38.6 ± 11.4 years, 55.2% male). The level of education of most participants was high school (47.4%) or university (42.1%).

Random allocation of participants resulted in no significant differences in the distribution of sociodemographic and medical characteristics of the sample between groups, except for BMI, where the mean was significantly lower in G2 than in the CG (*p* = 0.023). Random allocation also resulted in differences between groups in the type (*p* = 0.016) and location of dental procedures (*p* = 0.022). The mean time of dental visits was 32.4 ± 16.9 min, with no significant differences between groups (*p* = 0.348). A total of seven volunteer dentists participated in the study (mean age 36.57 ± 6.11 years, 57.1% male, with 85.7% general practitioners).

### 3.2. Microbiological Outcomes

From the initial sample of fifty-nine individuals, three were excluded from the microbiological analysis because the device was removed during the dental visit (two dentists (G2 and G3) asked to remove the device to allow completion of the dental procedure, and one patient (G2) did not tolerate the device and asked to have it removed). Thus, the microbiological analysis included 56 participants.

Table 2 summarizes the results of the microbiological evaluation in the different experimental groups, showing the mean Δ of CFU identified at each sampling point, as well as the overall mean Δ of CFU for each group. The use of the different device prototypes significantly reduced the contamination in the light reflector compared to the control group (01.46 ± 4.43 Δ CFU in G2 vs. 19.25 ± 36.50 Δ CFU in CG; *p* = 0. 028), as well as in the apron (09.11 ± 12.05 Δ CFU in G3 versus 21.14 ± 43.41 Δ CFU in GC; *p* = 0.044), and in the face shield (08.80 ± 32.70 Δ CFU in G1 versus 56.78 ± 76.64 Δ CFU in GC; *p* = 0.017).

The overall mean Δ of CFU was not significantly different between groups (*p* = 0.392), but a trend towards lower contamination was observed in all groups using the device compared to the CG. The results of all RT-PCR COVID-19 tests performed on the patients and dentists were negative.

### 3.3. Ergonomic Outcomes

Table 3 shows the ergonomic results for the different experimental groups, considering both the patients’ and the dentists’ perspectives. The use of the different prototypes of the device was well accepted by the patients, as most of them (77.8%) considered the use as “good” or “very good”. The mean score for the Likert-scale question to patients was slightly lower in G3 (3.87 ± 0.64) than in G1 (4.07 ± 0.59) and G2 (4.00 ± 0.88), but no statistical difference was observed between the groups (*p* = 0.651).

The use of the different prototypes of the device significantly increased the dentists’ “feeling of safety and protection” (*p* < 0.001), especially in G1 (4.80 ± 0.41) compared to the CG (3.21 ± 1.12). On the other hand, from the dentists’ perspective, the use of the prototype device significantly decreased ergonomics in all domains.

In terms of “clinical visualization”, both prototypes without the aspiration system (G2: 2.53 ± 0.83; and G3: 2.47 ± 0.91) showed inferior performance compared to the CG (4.14 ± 1.10). The “clinical visualization” of the prototype with the aspiration system (G1: 3.20 ± 1.01) was not statistically different from the CG. Similarly, the ergonomic domain “ease of performing dental procedures” was significantly impacted by the use of the prototypes without the aspiration system (G2: 2.40 ± 0.91; and G3: 2.53 ± 0.91) compared to the CG (4.71 ± 0.73). G1 (3.60 ± 0.83) with the aspiration system did not differ from the CG in terms of “ease of performing dental procedures”. All the other ergonomic domains “physical access to the patient”, “agility to move”, and “space to work” were negatively affected by the use of the different prototypes of the device compared to the CG (*p* < 0.001).

Qualitative results from the open-ended question to both patients and dentists revealed the following summarized perceptions regarding the use of the device: “the use of the devices increased my feeling of protection during the dental visit”; “the acrylic device (G1 and G2) resulted in increased difficulty in accessing the patient and lack of space to work”; and “the PVC device (G3) is fragile and resulted in particular difficulty in visualizing the operative field and little space to work”.

## 4. Discussion

This study was designed and conducted during a challenging period for health services, particularly in the field of dentistry. In the initial phase of the global response to the SARS-CoV-2 pandemic, the prevailing circumstances were characterized by a lack of clarity regarding the infection and transmission dynamics of the virus. At that juncture, health protocols advised minimizing the occurrence of medical and dental procedures that generated aerosols. To the best of our knowledge, this study represents one of the earliest investigations to assess the efficacy of protective barriers in a clinical setting, confronting the tangible challenges and complexities inherent to a public dental service.

The present randomized clinical trial was innovative in testing different prototypes of a device that has been demonstrated to reduce microbial contamination during dental care, particularly at the point of care where personal protective equipment (PPE) was worn by dental professionals. Moreover, the present results confirm previous findings from a laboratory study published by the same research group using one of the prototypes [16]. It is noteworthy that the tested device may represent a valuable supplemental resource to infection prevention and control in dentistry, not only during the COVID-19 pandemic, but also for the management of future infectious diseases and epidemics.

Present findings suggested that the reduction in aerosol dispersion and contamination occurred with all three tested prototypes, mainly close to the operator area. However, the device with an aspiration system demonstrated a lower mean CFU count compared to the other prototypes. These results are in agreement with the recommendations set forth by the Food and Drugs Administration (FDA) [22], which highlights the use of negative pressure chambers as an important item in barrier devices. Moreover, present findings are in accordance with results found in the study by Comisi [17], who observed a reduction in aerosol dispersion, mainly in the operator’s face shield area and especially when aspiration was integrated into the setup device.

The results of this study indicate that distant areas were less contaminated, including those from the CG. These findings are corroborated by a similar study that demonstrated the dilution effect of the water spray from dental instruments and the positive effect of dental suction [23]. Furthermore, opening windows was an additional recommended measure at the time of the experiment [24,25], and may have contributed to contamination in some areas, such as the bench, table, and wall.

In the present study, the use of the device was well accepted by patients and resulted in an increased feeling of safety and protection among dentists. Conversely, the use of the device resulted in a notable decline in ergonomics during dental procedures. Interestingly, in G1 where the aspiration system was employed, the ergonomic impacts were reduced in certain domains, particularly in relation to the “clinical visualization” and the “ease to performing dental procedures”, with no statistical differences from the CG. It may be inferred that the use of a negative pressure system may serve to reduce the precipitation of particles on the internal walls of the acrylic chamber, thereby favoring the visualization of the operatory field by dentists. The remaining ergonomic domains, namely “physical access to the patient”, “agility to move”, and “space to work” were adversely affected by using different prototypes of the device, compared to the CG. Nevertheless, these ergonomic impacts did not result in an increase in working time, as the average time remained similar in all the test and control groups. Still, it is clear that there is room for improvement in the ergonomic aspects of the equipment, such as advances in its design, with a possible reduction in its dimensions, or even the use of other materials that allow thinner walls and a lower weight, or materials that facilitate the disinfection process between consultations.

The present study has limitations that need to be disclosed. The difficulty in obtaining supplies such as PPE and laboratory equipment during the first months of the pandemic delayed the start of the study. In addition, the difficulty in recruiting dentists and patients requiring aerosol-generating procedures was another limitation, as these procedures were avoided at the time of the experiment. Moreover, at the time of the study, all dentists from the CMOBM service (*n* = 9) were invited to participate in the study and all dentists who agreed to participate were included (*n* = 7). However, prior sample-size calculation for dentists was not feasible in this scenario and we acknowledge this as a potential limitation of this study. On the other hand, we believe that the inclusion of most dentists from this public dental service reflects the real scenario of dental practice at that time. Furthermore, we acknowledge that the present study may not reflect the characteristics of all dental specialties among the participating dentists, considering the limited sample of dentists.

In addition, participants were randomly allocated to the different study groups according to the CONSORT guidelines to avoid selection bias. This strategy allowed for the distribution of most sociodemographic and medical variables (age, sex, scholarship, smoking, CVD, hypertension, and diabetes) meaning they did not differ between the study groups, which is a methodological strength. The results in Table 1 show that even after taking into account the randomization process, a statistical difference in the distribution of patients’ BMI between the CG (27.6 ± 2.8) and G2 (23.8 ± 1.9) was detected. This difference was not considered clinically relevant as there were no statistical differences in BMI between the three experimental groups (G1, G2, and G3). BMI was analysed as a possible confounding variable in relation to the ergonomic analysis, as it could be hypothesized that patients with a higher BMI would have worse ergonomic results when using the devices. Table 1 also shows a difference between groups in the distribution of the type and location of procedures performed during dental visits (a higher frequency of restorations in the CG and a higher frequency of prophylaxis of all teeth in the experimental groups). Again, this difference was present even after random allocation of participants. However, we believe that this may not represent an important bias, since the procedures that most generate aerosol dispersion and potential contamination (all-tooth prophylaxis) were more frequently performed in the experimental groups that used the device (so a higher contamination would be expected in these groups compared to the control group); however, the microbiological results showed a statistically lower contamination in the groups that used the device compared to the control group, even in the presence of these differences related to the type and location of the procedures. Another inherent limitation of this study is the limited sample size and representativeness, as the patient population was restricted to a specific region and demographic group. Further studies using the device in different dental services and larger populations are strongly encouraged.

Finally, the use of preprocedural mouthwashes has been suggested to reduce microbial load prior to dental procedures, and it was recently demonstrated to be effective in reducing the number of respiratory pathogens present during dental aerosol-generating treatment [26]. However, when this clinical trial was carried out, there was no established mouthwash protocol in the service, so no preprocedural oral disinfection was performed. It is possible that the addition of a preprocedural mouth-washing protocol in combination with the device tested would reduce the burden of microbial spread.

It is important to emphasize that the conduct of this study during the pandemic period was only possible thanks to a partnership between the university and a public military institution whose health and dental services remained operational to serve professionals who were at the forefront of the fight against SARS-CoV-2. Finally, although the microbiological results of using the device were favorable, it is clear that further ergonomic improvements to the device are needed to enable its large-scale use in dental practice.

## 5. Conclusions

The prototypes of the device tested reduced microbial contamination during dental procedures, particularly on the dentist’s PPE (face shield and apron). The use of the devices was well accepted by the patients and resulted in an increased feeling of safety and protection by the dentists but significantly reduced the ergonomics of the dental practice. The tested device can be an additional resource for infection prevention and control in dentistry not only during the COVID-19 pandemic, but also for the control of future infectious diseases and epidemics.

## 6. Patents

The author R.T.-F. owns a pending patent for the device described in the study.

## Figures and Tables

**Figure 1 dentistry-13-00054-f001:**
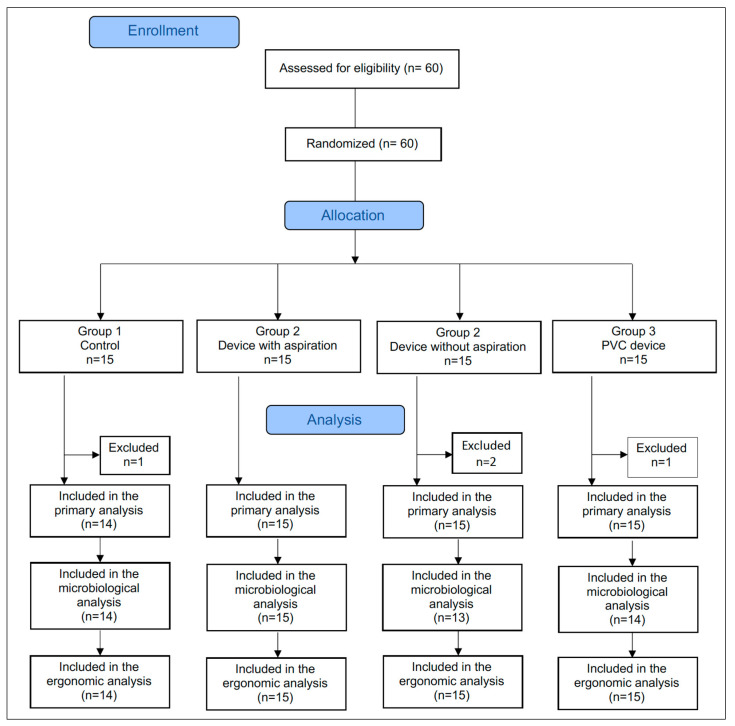
CONSORT flowchart of sample selection and allocation.

**Figure 2 dentistry-13-00054-f002:**
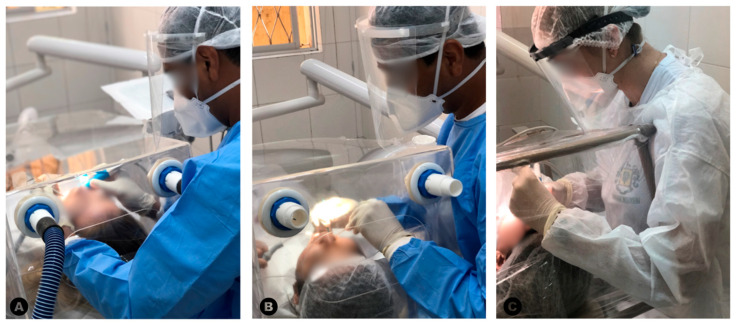
Representative images of the dental procedures performed using the prototypes in G1 (**A**), G2 (**B**), and G3 (**C**).

**Table 1 dentistry-13-00054-t001:** Sociodemographic and medical characteristics of participants, and variables related to the operative procedures in the different groups (*n* = 59). Results are expressed as *n* (%) or mean ± standard deviations.

	CG (Control) *n* = 14	G1 (Device + Aspiration) *n* = 15	G2 (Device Only) *n* = 15	G3 (PVC) *n* = 15	*p*-Value *	Total Sample
*Sociodemographic variables*						
**Age** (*n* = 55)	41.7 ± 13.8	35.3 ± 9.4	37.9 ± 12.3	40.0 ± 10.7	0.624	38.6 ± 11.4
**Sex** (*n* = 58)					0.979	
Female	06 (46.2)	07 (46.7)	06 (40.0)	07 (46.7)		26 (44.8)
Male	07 (53.8)	08 (53.3)	09 (60.0)	08 (53.3)		32 (55.2)
**Scholar** (*n* = 57)					0.525	
Elementary	02 (15.4)	01 (06.7)	00 (00.0)	03 (20.0)		06 (10.5)
High school	06 (46.2)	07 (46.7)	06 (42.9)	08 (53.3)		27 (47.4)
University	05 (38.5)	07 (46.7)	08 (57.1)	04 (26.7)		24 (42.1)
*Medical variables*						
**BMI** (*n* = 50)	27.6 ± 2.8 ^a^	25.1 ± 3.2 ^ab^	23.8 ± 1.9 ^b^	26.2 ±4.5 ^ab^	**0.023**	25.6 ± 3.4
**Smoke** (N = 57)					0.105	
Yes	01 (07.7)	04 (26.7)	00 (00.0)	01 (06.7)		06 (10.5)
No	12 (92.3)	11 (73.3)	14 (100)	14 (93.3)		51 (89.5)
**CVD** (*n* = 57)					0.601	
Yes	00 (00.0)	01 (06.7)	00 (00.0)	01 (06.7)		02 (03.5)
No	13 (100)	14 (93.3)	14 (100)	14 (93.3)		55 (96.5)
**Hypertension** (*n* = 57)					0.921	
Yes	02 (15.4)	02 (13.3)	01 (07.1)	02 (13.3)		07 (12.3)
No	11 (84.6)	13 (86.7)	13 (92.9)	13 (86.7)		50 (87.7)
**Diabetes** (*n* = 57)					0.415	
Yes	00 (00.0)	00 (00.0)	00 (00.0)	01 (06.7)		01 (01.8)
No	13 (100)	15 (100)	14 (100)	14 (93.3)		56 (98.2)
*Variables related to the procedures*						
**Type (***n* = 59)					**0.016**	
Restoration	11 (78.6)	02 (13.3)	08 (53.3)	06 (40.0)		27 (45.8)
Coron access	01 (07.1)	01 (06.7)	00 (00.0)	02 (13.3)		04 (06.8)
Prophylaxis	02 (14.3)	12 (80.0)	07 (46.7)	07 (46.7)		28 (47.5)
**Local** (*n* = 59)					**0.022**	
Upper Inc/Cusp	03 (21.4)	00 (00.0)	01 (06.7)	05 (33.3)		09 (15.3)
Inf Inc/Cusp	00 (00.0)	01 (06.7)	00 (00.0)	00 (00.0)		01 (01.7)
Upper Bic/Molar	06 (42.9)	02 (13.3)	04 (26.7)	02 (13.3)		14 (23.7)
Inf Bic/Molar	03 (21.4)	00 (00.0)	03 (20.0)	01 (06.7)		07 (11.9)
All teeth	02 (14.3)	12 (80.0)	07 (46.7)	07 (46.7)		28 (47.5)
**Time** (min) (*n* = 58)	34.3 ± 13.2	27.0 ± 9.2	28.8 ± 12.9	39.0 ± 25.7	0.348	32.4 ± 16.9

* *p*-value for the Kruskal–Wallis test for independent samples, with pairwise comparisons (continuous variables) or the Chi-Square (categorical variables); bold values = statistical significant differences. ^ab^ Different letters indicate a significant difference between the groups (*p* < 0.05).

**Table 2 dentistry-13-00054-t002:** Results of the microbiological evaluation in the different experimental groups. Mean Δ of CFU identified at each sampling point, as well as the total mean Δ of CFU for each group. Results are expressed as mean ± standard deviations.

Local	CG (Control) *n* = 14	G1 (Device + Aspiration) *n* = 15	G2 (Device Only) *n* = 13	G3 (PVC) *n* = 14	*p*-Value *
Δ Light reflector	19.25 ± 36.50 ^ab^	12.03 ± 19.11 ^a^	−08.07 ± 38.22 ^b^	15.18 ± 36.67 ^ab^	**0.028**
Δ Backrest	27.32 ± 56.57	20.57 ± 54.60	11.58 ± 30.54	01.60 ± 02.75	0.978
Δ Clinical table	06.89 ± 28.81	04.43 ± 14.48	01.23 ± 01.85	00.64 ± 01.18	0.926
Δ Apron	21.14 ± 43.41	51.50 ± 65.92	43.88 ± 65.96	09.11 ± 12.05	**0.044**
Δ Mask	04.39 ± 08.26	3.43 ± 9.37	−02.70 ± 06.34	−01.39 ± 06.37	0.174
Δ Faceshield	56.78 ± 76.64 ^a^	08.80 ± 32.70 ^b^	01.34 ± 08.64 ^ab^	02.75 ± 07.17 ^ab^	**0.017**
Δ Cap	04.93 ± 14.99	−01.80 ± 28.46	02.11 ± 06.26	01.71 ± 06.43	0.686
Δ Bib	80.05 ± 116.21	55.93 ± 67.51	70.07 ± 113.89	92.82 ± 101.84	0.889
Δ Bench	02.61 ± 07.93	03.93 ± 04.57	−14.30 ± 49.03	00.86 ± 03.92	0.475
Δ Wall	−12.71 ± 33.88	00.43 ± 01.52	−19.57 ± 46.80	−00.32 ± 01.10	0.453
Δ Overall	21.11 ± 26.32	15.93 ± 10.48	08.46 ± 19.43	12.30 ± 10.15	0.392

* *p*-value for the Kruskal–Wallis test with pairwise comparisons (continuous variables); bold values = statistical significant differences. ^ab^ Different letters indicate significant differences between groups (*p* < 0.05).

**Table 3 dentistry-13-00054-t003:** Results of the ergonomic impacts of the different prototypes of the device from the patient’s and dentist’s perspective. Results are expressed as N (%) or mean ± standard deviations.

			CG (Control) *n* = 14	G1 (Device + Aspiration) *n* = 15	G2 (Device Only) *n* = 15	G3 (PVC) *n* = 15	*p*-Value *
**Evaluation** **patients**	Very bad		N/A	00 (00.0)	00 (00.0)	00 (00.0)	0.648
	Bad			00 (00.0)	01 (07.1)	00 (00.0)	
	Regular			02 (13.3)	02 (14.3)	04 (26.7)	
	Good			10 (66.7)	07 (50.0)	09 (60.0)	
	Very good			03 (20.0)	04 (28.6)	02 (13.3)	
	Mean ± s.d.		N/A	4.07 ± 0.59	4.00 ± (0.88)	3.87 ± (0.64)	0.651
**Evaluation professionals**	Sensation of security and protection	Very bad	01 (07.1)	00 (00.0)	00 (00.0)	00 (00.0)	**0.001**
		Bad	02 (14.3)	00 (00.0)	00 (00.0)	00 (00.0)	
		Regular	06 (42.9)	00 (00.0)	00 (00.0)	05 (33.3)	
		Good	03 (21.4)	03 (20.0)	07 (46.7)	07 (46.7)	
		Very good	02 (14.3)	12 (80.0)	08 (53.3)	03 (20.0)	
		Mean ± s.d.	3.21 ± 1.12 ^a^	4.80 ± 0.41 ^b^	4.53 ± 0.52 ^bc^	3.87 ± 0.74 ^ac^	**<0.001**
	Clinical visualization	Very bad	00 (00.0)	02 (13.3)	01 (06.7)	02 (13.3)	**<0.001**
		Bad	02 (14.3)	00 (00.0)	07 (46.7)	06 (40.0)	
		Regular	01 (07.1)	06 (40.0)	05 (33.3)	05 (33.3)	
		Good	04 (28.6)	07 (46.7)	02 (13.3)	02 (13.3)	
		Very good	07 (50.0)	00 (00.0)	00 (00.0)	00 (00.0)	
		Mean ± s.d.	4.14 ± 1.10 ^a^	3.20 ± 1.01 ^ab^	2.53 ± 0.83 ^b^	2.47 ± 0.91 ^b^	**<0.001**
	Physical access to the patient	Very bad	00 (00.0)	00 (00.0)	06 (40.0)	03 (20.0)	**<0.001**
		Bad	00 (00.0)	02 (13.3)	02 (13.3)	04 (26.7)	
		Regular	00 (00.0)	08 (53.3)	05 (33.3)	06 (40.0)	
		Good	02 (14.3)	05 (33.3)	02 (13.3)	02 (13.3)	
		Very good	12 (85.7)	00 (00.0)	00 (00.0)	00 (00.0)	
		Mean ± s.d.	4.86 ± 0.36 ^a^	3.20 ± 0.68 ^b^	2.20 ± 1.15 ^b^	2.47 ± 0.99 ^b^	**<0.001**
	Ease of performing procedure	Very bad	00 (00.0)	00 (00.0)	03 (20.0)	01 (06.7)	**<0.001**
		Bad	00 (00.0)	02 (13.3)	04 (26.7)	08 (53.3)	
		Regular	02 (14.3)	03 (20.0)	07 (46.7)	03 (20.0)	
		Good	00 (00.0)	09 (60.0)	01 (06.7)	03 (20.0)	
		Very good	12 (85.7)	01 (06.7)	00 (00.0)	00 (00.0)	
		Mean ± s.d.	4.71 ± 0.73 ^a^	3.60 ± 0.83 ^ab^	2.40 ± 0.91 ^b^	2.53 ± 0.91 ^b^	**<0.001**
	Agility to move	Very bad	00 (00.0)	01 (06.7)	02 (13.3)	01 (06.7)	**<0.001**
		Bad	00 (00.0)	02 (13.3)	07 (46.7)	07 (46.7)	
		Regular	00 (00.0)	06 (40.0)	03 (20.0)	07 (46.7)	
		Good	04 (28.6)	06 (40.0)	03 (20.0)	00 (00.0)	
		Very good	10 (71.4)	00 (00.0)	00 (00.0)	00 (00.0)	
		Mean ± s.d.	4.71 ± 0.47 ^a^	3.13 ± 0.91 ^b^	2.47 ± 0.99 ^b^	2.40 ± 0.63 ^b^	**<0.001**
	Space to work	Very bad	00 (00.0)	01 (06.7)	00 (00.0)	01 (06.7)	**<0.001**
		Bad	00 (00.0)	01 (06.7)	05 (33.3)	03 (20.0)	
		Regular	00 (00.0)	07 (46.7)	07 (46.7)	08 (53.3)	
		Good	04 (28.6)	06 (40.0)	03 (20.0)	02 (13.3)	
		Very good	10 (71.4)	00 (00.0)	00 (00.0)	01 (06.7)	
		Mean ± s.d.	4.71 ± 0.47 ^a^	3.20 ± 0.86 ^b^	2.87 ± 0.74 ^b^	2.93 ± 0.96 ^b^	**<0.001**

* *p*-value for the Kruskal–Wallis test with pairwise comparisons (continuous variables) or the Chi-Square (categorical variables); bold values = statistical significant differences. ^abc^ Different letters indicate significant differences between groups (*p* < 0.05).

## Data Availability

The original contributions presented in this study are included in the Appendix A. Further inquiries can be directed to the corresponding author.

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
