# Peer review of "Microbiological and Ergonomic Effects of Three Prototypes of a Device to Reduce Aerosol Dispersion in Dental Care During the COVID-19 Pandemic: A Randomized Controlled Clinical Trial"

_dentistry, 2025, doi:10.3390/dj13020054_

Round 1

Reviewer 1 Report

Comments and Suggestions for Authors

This study is a meaningful clinical evaluation of the efficacy of device prototypes developed to reduce microbial contamination during dental procedures. However, the following revisions are necessary:

1. The Methods section in the Abstract lacks connection to the Results section. Please include details about the three device prototypes in the Methods section of the Abstract.

2. In the Introduction, swapping the second and third paragraphs and adjusting the sentence order within the combined paragraph would create a more cohesive narrative.

3. The differences between G1 and G2 are clear and understandable. However, the differences between G2 and G3 are less distinct. Please provide additional explanation to clarify these differences.

4. Add information about the manufacturer of the agar medium used in the study.

5. The study appropriately calculated the sample size for patients. However, there is no discussion on the rationale or justification for the sample size of participating dentists. If determining an appropriate sample size for dentists is challenging, please acknowledge this as a limitation.

6. The study randomized patients into CG, G1, G2, and G3 groups. What is the significance of the comparisons made in Table 1?

7. Include a limitation regarding the limited representativeness of the study due to the patient population being restricted to a specific region and demographic group.

8. The study did not reflect the characteristics of specific dental specialties among the participating dentists. This should also be acknowledged as a limitation.

Author Response

RESPONSES TO THE REVIEWER 1:

Referee(s) comments to authors:

Referee:1

 This study is a meaningful clinical evaluation of the efficacy of device prototypes developed to reduce microbial contamination during dental procedures. However, the following revisions are necessary:

  1. The Methods section in the Abstract lacks connection to the Results section. Please include details about the three device prototypes in the Methods section of the Abstract.

Response 1: We are grateful for your careful critical review and important insights. We are happy you realized the clinical implication of our findings. According to your suggestion, additional details about the three device prototypes where included in the Methods section of the Abstract. The following sentence was included (lines 22-24): “The device prototypes consisted of a rigid translucent acrylic structure (G1 and G2) or a rigid PVC tube structure surrounded by layers of translucent flexible PVC films (G3), adjusted to the dental chair, involving the patient’s head, neck and chest.”

  1. In the Introduction, swapping the second and third paragraphs and adjusting the sentence order within the combined paragraph would create a more cohesive narrative.

Response 2: We thank you for your suggestion and have carefully analysed it. We respectfully suggest that the current order of the paragraphs is appropriate, as the second paragraph refers to a broader context (aerosol generation during health care procedures and previous protective devices proposed in the medical field), while the third paragraph focuses specifically on the problem and previous initiatives proposed to reduce contamination during dental care. 

  1. The differences between G1 and G2 are clear and understandable. However, the differences between G2 and G3 are less distinct. Please provide additional explanation to clarify these differences.

Response 3: We are happy to have the opportunity to clarify this issue. Please observe that in the Methods section we exposed the difference between the three different prototypes (lines 112-125). The device used in G1 was extensively described in a previous publication of our research group (please see details in: Teichert-Filho, R.; Baldasso, C.N.; Campos, M.M.; Gomes, M.S. Protective device to reduce aerosol dispersion in dental clinics during the COVID-19 pandemic. Int Endod J 2020, 53, 1588-1597, doi:10.1111/iej.13373). Thus, in G1 and G2 the device consisted of the same rigid translucent acrylic structure (methyl polymethacrylate), adjusted to the dental chair, involving the patient’s head, neck and chest. In G1, there is also a piping

system to generate negative pressure, for aspiration and filtering of the air inside the device chamber. On the other hand, G3 consisted of a simplified polyvinyl chloride (PVC) device. This device in G3 consisted of a rigid PVC tube structure surrounded by layers of translucent flexible PVC films. As in G1 and G2, the device used in G3 was adapted to the dentist’s chair, involving the patient's head and chest. In all G1, G2 and G3 groups, the operator worked through small holes in the acrylic structure (G1 and G2) or in the translucent flexible PVC films (G3), to reduce contact with the microparticles arising from aerosols during dental procedures. Please note that the main difference between G2 and G3 is the material used in the structure: G2 – rigid translucent acrylic structure (methyl polymethacrylate); and G3 - rigid PVC tube structure surrounded by layers of translucent flexible PVC films. You may find a picture showing these differences in Figure 2 of the manuscript.

  1. Add information about the manufacturer of the agar medium used in the study.

Response 4: The information was included accordingly (lines 144-145): “After preparation of serial dilutions, viable bacteria were cultured in Petri plates containing blood agar medium (Blood-Agar-TSA-20mL-PL 90X15-PC 10PL, Laborclin, Pin-hais/PR, Brazil) and incubated at 37°C for 24 h.”

  1. The study appropriately calculated the sample size for patients. However, there is no discussion on the rationale or justification for the sample size of participating dentists. If determining an appropriate sample size for dentists is challenging, please acknowledge this as a limitation.

Response 5: We are grateful for your comment. This study was conducted during the apex of the pandemic period, from August 2020 to July 2021. At that time, all dentists from the CMOBM service (N=9) were invited to participate in the study, and all dentists who agree to participate were included (N=7). Thus, a previous sample size calculation for dentists was not feasible in that scenario, and we acknowledge it as a possible limitation of this study. On the other hand, we believe that including most dentists from that public dental service reflects the real scenario of dental practice at that time. A sentence was included in the Discussion section (lines 332-337): “Moreover, at the time of the study, all dentists from the CMOBM service (N=9) were invited to participate in the study and all dentists who agreed to participate were included (N=7). However, prior sample size calculation for dentists was not feasible in this scenario and we acknowledge this as a potential limitation of this study. On the other hand, we believe that the inclusion of most dentists from this public dental service reflects the real scenario of dental practice at that time.”

  1. The study randomized patients into CG, G1, G2, and G3 groups. What is the significance of the comparisons made in Table 1?

Response 6: Table 1 shows the sociodemographic and medical characteristics of participants, and variables related to the operative procedures in the different groups (CG, G1, G2 and G3). Yes, participants were randomly allocated in the different study groups, according to the CONSORT guidelines, in order to avoid selection bias. This strategy allowed that the distribution of most sociodemographic and medical variables (age, sex, scholar, smoke, CVD, hypertension, diabetes) were not different among the study groups, which is a methodological strength. Results in Table 1 also shows that, even caring out the randomization process, a statistical difference in the distribution of patient’s BMI between CG (27.6±2.8) and G2 (23.8±1.9) was detected. This difference was not considered clinically relevant, since there was no statistical differences in the BMI among the three experimental groups using the divide (G1, G2 and G3). BMI was analysed as a possible confounding variable related to the ergonomic analysis, since it may be hypothesized that patients with a higher BMI would experience worse ergonomic outcomes using the devices. Moreover, Table 1 also revealed a difference among groups in relation to the distributions of the type and local of the procedures performed during dental visits (a higher frequency of restorations in the CG and a higher frequency of prophylaxis of all teeth in the experimental groups). Again, this difference was present even after the random allocation of participants. However, we believe that this may not represent an important bias, since the procedures that most generate aerosol dispersion and potential contamination (all teeth prophylaxis) were more frequently performed in the experimental groups that used the device (so, a higher contamination would be expected in these groups compared to the control group); however, the microbiological results revealed a statistically lower contamination in the groups using the device, compared to the control group, even in the presence of these differences related to the type and local of the procedures.

Thus, a new paragraph was included in the discussion section (lines 341-362): “In addition, participants were randomly allocated to the different study groups according to CONSORT guidelines to avoid selection bias. This strategy allowed that the distribution of most socio-demographic and medical variables (age, sex, scholarship, smoking, CVD, hypertension, diabetes) did not differ between the study groups, which is a methodological strength. The results in Table 1 show that even after taking into account the randomization process, a statistical difference in the distribution of patients' BMI between CG (27.6±2.8) and G2 (23.8±1.9) was detected. This difference was not considered clinically relevant as there were no statistical differences in BMI between the three experimental groups (G1, G2 and G3). BMI was analysed as a possible confounding variable in relation to the ergonomic analysis, as it could be hypothesized that patients with a higher BMI would have worse ergonomic results when using the devices. Table 1 also shows a difference between groups in the distribution of the type and location of procedures performed during dental visits (a higher frequency of restorations in the CG and a higher frequency of prophylaxis of all teeth in the experimental groups). Again, this difference was present even after random allocation of participants. However, we believe that this may not represent an important bias, since the procedures that most generate aerosol dispersion and potential contamination (all-tooth prophylaxis) were more frequently performed in the experimental groups that used the device (so a higher contamination would be expected in these groups compared to the control group); however, the microbiological results showed a statistically lower contamination in the groups that used the device compared to the control group, even in the presence of these differences related to the type and location of the procedures.”.

  1. Include a limitation regarding the limited representativeness of the study due to the patient population being restricted to a specific region and demographic group.

Response 7: We agree. The following sentence was included in the Discussion section (lines 362-365): “Another inherent limitation of this study is the limited sample size and representativeness, as the patient population was restricted to a specific region and demographic group. Further studies using the device in different dental services and larger populations are strongly encouraged.”.

  1. The study did not reflect the characteristics of specific dental specialties among the participating dentists. This should also be acknowledged as a limitation.

Response 8: We agree and included the following sentence in the Discussion section (lines 338-340): “Furthermore, we acknowledge that the present study may not reflect the characteristics of all dental specialties among the participating dentists, considering the limited sample of dentists.”.

We appreciate your valuable review. We have tried to attend to all the raised questions and have made the alterations throughout the manuscript, as requested. We hope our efforts have improved the quality of our study and you may now find it satisfactory.

Reviewer 2 Report

Comments and Suggestions for Authors

Thank you for the opportunity to review this manuscript. 

I have some recommendations:

-to be inline with contemporary language change "infection control" to "infection prevention and control"

- provide a definition of "aerosol" and the relevance to the dispersal of all potential pathogens plus transmission of respiratory pathogens with particular reference to COVID-19

-the small sample size needs to be recognised as a limitation

Author Response

RESPONSES TO THE REVIEWER 2:

Referee(s) comments to authors:

Referee # 2

Thank you for the opportunity to review this manuscript.

I have some recommendations:

-to be inline with contemporary language change "infection control" to "infection prevention and control"

Response 1: We are grateful for your review and important insights. We agree and have changed the terms accordingly all over the manuscript.

- provide a definition of "aerosol" and the relevance to the dispersal of all potential pathogens plus transmission of respiratory pathogens with particular reference to COVID-19

Response 2: We included the following sentences in the Introduction section (lines 52-57): “By definition, an aerosol is a suspension of fine solid particles or liquid droplets in a gas. Aerosols can vary greatly in size, with the term often used to describe particles smaller than 100 µm, which can remain airborne for extended periods of time [8]. Aerosol particles <10 µm can penetrate into the lower respiratory tract, facilitating the transmission of respiratory pathogens such as the severe acute respiratory syndrome coronavirus 2 (SARS-CoV-2).”.

-the small sample size needs to be recognized as a limitation

Response 3: We agree and have included it in the discussion section (please see lines 362-365 and also 338-340).

We appreciate your valuable review. We have tried to attend to all the raised questions and have made the alterations throughout the manuscript, as requested. We hope our efforts have improved the quality of our study and you may now find it satisfactory.

Reviewer 3 Report

Comments and Suggestions for Authors

In the manuscript entitled “Microbiological and Ergonomic Effects of Three Prototypes of a Device to Reduce Aerosol Dispersion in Dental Care During the COVID-19 Pandemic: A Randomized Controlled Clinical Trial” the authors aimed to evaluate the microbiological efficacy and the ergonomic impact of three prototypes of a device to reduce aerosol dispersion during dental procedures. The writing is clear, but some sections could benefit from additional detail and context. Here are reported my suggestions to improve the quality of the manuscript:

ABSTRACT

_ Numerical details on microbiological reductions and ergonomic scores should be specified in the abstract.

INTRODUCTION

_ The Introduction could be expanded by presenting the recent advancements in aerosol management techniques or alternative device designs.

_ It is not clear how this study addressed the limitations of previous studies.

MATERIALS AND METHODS

_ The rationale for choosing the specific prototypes is not fully explained.

_The ergonomic questionnaire could be detailed further to improve reproducibility. For instance, the phrasing of Likert-scale questions is not provided.

RESULTS

_ The presentation of microbiological outcomes could benefit from more context on clinical significance (e.g., thresholds for meaningful contamination reduction).

_ I am not sure the style of the tables fit with that of MDPI templates.

DISCUSSION

Strengths:

_ The discussion does not propose specific strategies for improving device ergonomics or addressing the observed shortcomings.

_ Limitations are mentioned but could be expanded, such as the lack of long-term follow-up or larger sample sizes.

GENERAL

_ A flowchart summarizing the experimental process would enhance clarity.

Author Response

RESPONSES TO THE REVIEWER 3:

Referee(s) comments to authors:

Referee # 3

In the manuscript entitled “Microbiological and Ergonomic Effects of Three Prototypes of a Device to Reduce Aerosol Dispersion in Dental Care During the COVID-19 Pandemic: A Randomized Controlled Clinical Trial” the authors aimed to evaluate the microbiological efficacy and the ergonomic impact of three prototypes of a device to reduce aerosol dispersion during dental procedures. The writing is clear, but some sections could benefit from additional detail and context. Here are reported my suggestions to improve the quality of the manuscript:

ABSTRACT

_ Numerical details on microbiological reductions and ergonomic scores should be specified in the abstract.

Response 1: We are grateful for your careful critical review and important insights. Following your suggestion, additional details related to the microbiological reductions were included in the abstract. Due to limitations in the number of words in the abstract, ergonomic scores were not included in the abstract, but can be observed in details in Table 3 of the Results section.

INTRODUCTION

_ The Introduction could be expanded by presenting the recent advancements in aerosol management techniques or alternative device designs.

Response 2: Please note that in the introduction section are included references to alternative devices to manage aerosol dispersion, and new recent references were included (lines 62-66).

_ It is not clear how this study addressed the limitations of previous studies.

Response 3: To date, the microbiological efficacy of existing devices and their ergonomic impact in dental care have not been sufficiently tested in clinical settings, and randomized clinical trials in this field are scarce. Most previous studies were conducted under laboratory conditions and used simulated procedures to analyse the devices, which may not reflect the real dental care situations. Please note that we have addressed it in the last paragraph of the Introduction section, and the following sentences were included (lines 67-70): “To date, the microbiological efficacy of existing devices and their ergonomic impact in dental care have not been sufficiently tested in clinical settings. Most previous studies were conducted under laboratory conditions and used simulated procedures to analyze the devices, which may not reflect real-life dental care situations.”.

MATERIALS AND METHODS

_ The rationale for choosing the specific prototypes is not fully explained.

Response 4: Our research group have previously developed a device and published a laboratory analysis of the device used in G1 (please see details in: Teichert-Filho, R.; Baldasso, C.N.; Campos, M.M.; Gomes, M.S. Protective device to reduce aerosol dispersion in dental clinics during the COVID-19 pandemic. Int Endod J 2020, 53, 1588-1597, doi:10.1111/iej.13373). That device was not tested in the real-life dental care settings, so we run the present randomized clinical trial using the original device with aspiration (G1), without aspiration (G2), and also included a low-cost simplified device (G3) to evaluate their microbiological efficacy and ergonomic impacts during dental procedures.

_The ergonomic questionnaire could be detailed further to improve reproducibility. For instance, the phrasing of Likert-scale questions is not provided.

Response 5: We agree. The full questions of the questionnaire were included in the Methods section (lines 160-169).

RESULTS

_ The presentation of microbiological outcomes could benefit from more context on clinical significance (e.g., thresholds for meaningful contamination reduction).

Response 6: This is a very interesting point. We acknowledge that establishing a clinically relevant threshold for bacterial contamination reduction in the dental clinical settings would be remarkable. However, to the best of our knowledge, there is no scientific data to support a “cut-off point” related to the clinical relevance of a CFU count reduction. In the clinical scenario, especially at the time of the apex of the COVID pandemic when the study was carried out, we believe that any statistically significant contamination reduction should be considered clinically important.

_ I am not sure the style of the tables fit with that of MDPI templates.

Response 7: We used the MDPI templates to write our manuscript, and the tables followed its format.

DISCUSSION

Strengths:

_ The discussion does not propose specific strategies for improving device ergonomics or addressing the observed shortcomings.

Response 8: We agree, and following your suggestion we added the following sentence in the Discussion section (lines 323-327): “Still, it is clear that there is room for improvement in the ergonomic aspects of the equipment, such as advances in its design, with a possible reduction in its dimensions, or even the use of other materials that allow thinner walls and lower weight, or materials that facilitate the disinfection process between consultations.”.

_ Limitations are mentioned but could be expanded, such as the lack of long-term follow-up or larger sample sizes.

Response 9: We agree and included other limitations. Please see lines to 328-365.

GENERAL

_ A flowchart summarizing the experimental process would enhance clarity.

Response 10: Please note that in Figure 1 we have included the recommended CONSORT flowchart of sample selection and allocation. We respectfully believe it is sufficient to clarify the experimental process.

We appreciate your valuable review. We have tried to attend to all the raised questions and have made the alterations throughout the manuscript, as requested. We hope our efforts have improved the quality of our study and you may now find it satisfactory.

Round 2

Reviewer 1 Report

Comments and Suggestions for Authors

Thank you for diligently addressing the reviewers' comments. This study holds significant value as a randomized controlled clinical trial evaluating the effectiveness of devices designed to reduce aerosol dispersion. The manuscript now meets the quality standards required for publication. 

Reviewer 3 Report

Comments and Suggestions for Authors

In their revised manuscript the authors gave interesting explanations and improved the manuscript to the best of their possibilities. I think that the paper could now be accepted for publication.